# Denoising Autoencoders for Unsupervised Anomaly Detection in Brain MRI

**Antanas Kascenas**[1,2]                                   ANTANAS.KASCENAS@MRE.MEDICAL.CANON

**Nicolas Pugeault**[2]                                      NICOLAS.PUGEAULT@GLASGOW.AC.UK

**Alison Q. O'Neil**[1,3]                                    ALISON.ONEIL@MRE.MEDICAL.CANON

[1] *Canon Medical Research Europe, Edinburgh, Scotland, UK*

[2] *University of Glasgow, Glasgow, Scotland, UK*

[3] *University of Edinburgh, Edinburgh, Scotland, UK*

**Editors:** Under Review for MIDL 2022

## Abstract

Pathological brain lesions exhibit diverse appearance in brain images, making it difficult to train supervised detection solutions due to the lack of comprehensive data and annotations. Thus, in this work we tackle unsupervised anomaly detection, using only healthy data for training with the aim of detecting unseen anomalies at test time. Many current approaches employ autoencoders with restrictive architectures (i.e. containing information bottlenecks) that tend to give poor reconstructions of not only the anomalous but also the normal parts of the brain. Instead, we investigate classical denoising autoencoder models that do not require bottlenecks and can employ skip connections to give high resolution fidelity. We design a simple noise generation method of upscaling low-resolution noise that enables high-quality reconstructions. We find that with appropriate noise generation, denoising autoencoder reconstruction errors generalize to hyperintense lesion segmentation and reach state of the art performance for unsupervised tumor detection in brain MRI data, beating more complex methods such as variational autoencoders. We believe this provides a strong and easy-to-implement baseline for further research into unsupervised anomaly detection.

**Keywords:** Anomaly detection, Unsupervised learning, Autoencoder, Denoising, MRI.

## 1. Introduction

Pathology detection is a popular task in medical imaging due to its wide range of possible applications. Supervised machine learning methods have shown promising results, however comprehensive supervised pathology detection methods require extensive and heterogeneous training sets which are costly to annotate and difficult to acquire. Conversely, unsupervised anomaly detection (UAD) methods require only identification of a healthy cohort of patients for training (therefore these methods are sometimes regarded as semi-supervised), after which they are applied to detect anomalous regions in test data.

Autoencoder deep learning methods have been commonly used for UAD in brain scans (Baur et al., 2021), relying on the assumption that data similar to that seen during training (healthy regions) will be reconstructed better than unseen (potentially anomalous) regions. Most autoencoder approaches are trained on healthy data using reconstruction error (e.g. mean squared error) as the main optimization objective. Anomaly scores are generated for each pixel often also using the reconstruction residuals. Models are then tested on data that

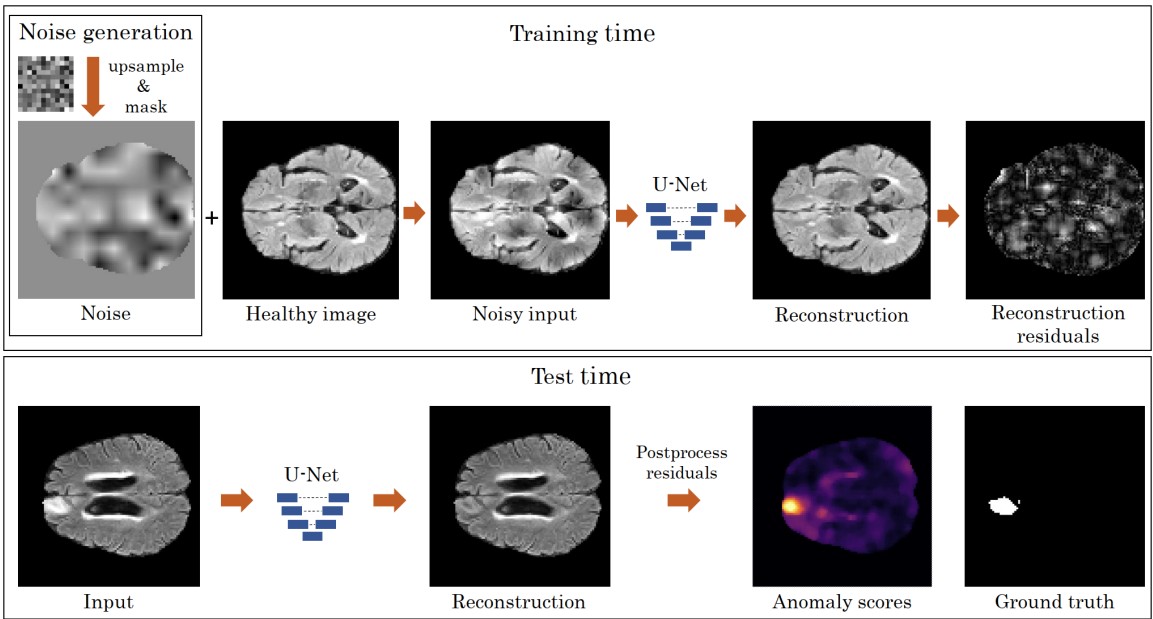

Figure 1: Our denoising autoencoder anomaly detection method. During training (top), noise is added to the foreground of the healthy image, and the network is trained to reconstruct the original image. At test time (bottom), the pixelwise post-processed reconstruction error is used as the anomaly score.

includes anomalies unseen during training. Since anomalies constitute out-of-distribution data, autoencoder models are expected to generalize poorly; thus, larger reconstruction errors are expected to indicate anomalous regions. However, it has been shown that in the main brain lesion datasets on which UAD methods are currently evaluated, a simple voxel intensity thresholding baseline with careful normalization and morphological post-processing performs better than most published autoencoder methods (Meissen et al., 2021). These results suggest that current methods do not effectively learn to detect intensity outliers.

In this paper, we explore the classic method of denoising autoencoders (DAEs). We find that DAEs produce better reconstructions than more popular autoencoder models with constrained architectures e.g., variational autoencoders, and that careful design of the injected noise allows models to be trained that are sensitive to subtle intensity changes and generalize to tumor localization in brain MRI scans. Our contributions are as follows:

1. We propose a DAE baseline for UAD that is simple to implement and outperforms an intensity thresholding baseline in brain tumor detection.

2. We show that DAEs with coarse noise patterns generalize significantly better than with traditional pixel/voxel-level noise.

3. We simplify and improve the intensity thresholding baseline; this remains a practical benchmark for methods trained on data that is not available to the evaluator.

## 2. Related Work

Many modifications to the standard autoencoder pipeline have been proposed.

Variational autoencoders (**VAEs**) Zimmerer et al. (2019); Zhou et al. (2020) constrain the latent bottleneck representation to follow a parameterized multivariate Gaussian distribution. Zimmerer et al. (2018) further add a context-encoding task and combine reconstruction error with density-based scoring to obtain the anomaly scores, while You et al. (2019) use an iterative gradient descent restoration process at test time in **restoration-VAE**, replacing the reconstruction error with a restoration error to estimate anomaly scores.

Architectural changes have also been proposed. Atlason et al. (2019); Baur et al. (2018) introduce **convolutional autoencoders** and higher capacity spatial bottlenecks instead of fully-connected (dense) bottlenecks to achieve better reconstruction. Chen and Konukoglu (2018) use **constrained autoencoders** to improve latent representation consistency in anomalous images at test time. **Bayesian skip-autoencoders** Baur et al. (2020) use skip connections with dropout to improve reconstruction and allow uncertainty to be measured via dropout stochasticity.

The UAD autoencoder framework of encoder-decoder components and reconstruction error for anomaly scores has featured in more complex approaches. Schlegl et al. (2019) train a generative adversarial network called **f-AnoGAN** which reuses the generator and discriminator to train an autoencoder with both reconstruction and adversarial losses for the anomaly detection task. Pinaya et al. (2021) combine a vector quantized VAE (**VQ-VAE**) to encode an image with a transformer model to resample low-likelihood latent variables in order to produce reconstructions with fewer reproduced anomalies.

Baur et al. (2021) have performed an evaluation of some of the most common autoencoder methods for anomaly detection in brain MRI finding restoration-VAE (You et al., 2019) and f-AnoGAN (Schlegl et al., 2019) to be among the best. However, more recently Meissen et al. (2021) have shown that most autoencoder-based UAD methods can be outperformed by a simple baseline. By using only the FLAIR modality and performing histogram equalization preprocessing they show that a simple thresholding-based method without any training can detect hyperintense brain tumor and multiple sclerosis lesions better than most UAD approaches that require healthy data to train.

The above evaluations omitted consideration of DAEs. DAEs have been applied as pretraining in brain lesion detection with limited labels and simple novelty detection using patch-based masking (Alex et al., 2017), and as a baseline using image-level Gaussian noise (Zimmerer et al., 2018). However, to our knowledge no works have investigated the effects of noise coarseness and intensity or achieved competitive results on the UAD task.

## 3. Method

We implement a simple denoising deep autoencoder neural network and use reconstruction error to detect and localize anomalies at test time. The training and test pipelines are visualized in Figure 1. Below we describe each part of the system in more detail.

**Network architecture**  We use a U-Net (Ronneberger et al., 2015) style architecture with skip connections which enables significantly better image reconstructions compared to bottleneck architectures such as the VAE (see Figure 2). However, any dense prediction (e.g.

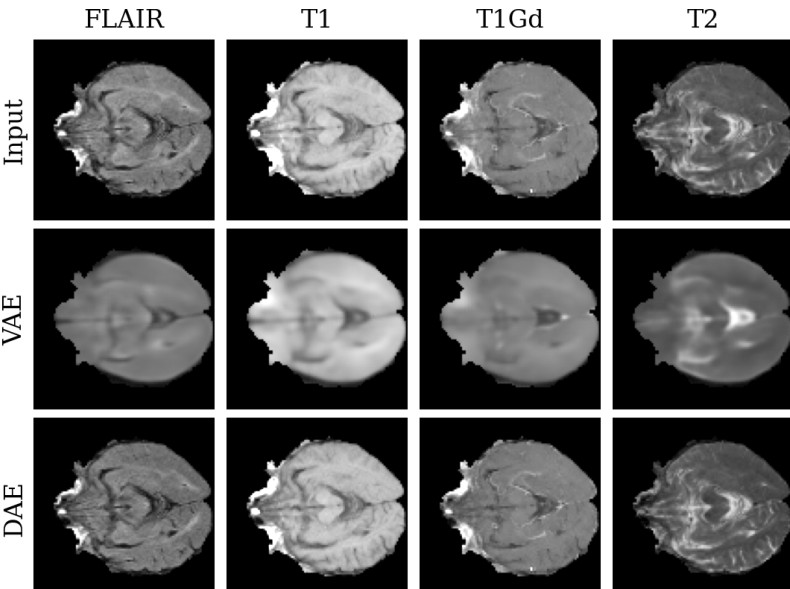

Figure 2: Sample healthy brain reconstructions from VAE and DAE models. The DAE gives more precise reconstructions. The VAE reconstruction quality could be improved by increasing bottleneck dimensionality, however this would negatively impact anomaly detection performance.

segmentation) neural network architecture can be easily repurposed for DAEs. Detailed description of the network architecture and training procedure can be found in Section 4.3 and Appendix A.

**Noise generation**  Randomly generated noise is added to each input image and the DAE is tasked with removing the noise and reconstructing the original input. DAEs perform denoising by learning to distinguish between healthy brain image patterns and random noise patterns. Thus, noise generation is essential for successful anomaly detection at test time. We generate coarse intensity noise by sampling random pixelwise Gaussian noise at a low resolution and bilinearly upsampling it to the input resolution. We then randomly translate the generated noise to avoid consistent upsampling patterns. Noise is added to the input foreground i.e. pixels with values above 0 (background pixels outside of the scan acquisition region are zero-valued). See Appendix B for examples of generated noise.

**Inference and post-processing**  The DAE is used to localize anomalies by calculating pixelwise anomaly scores $A(x)$ using $M = 4$ modalities of image $x$, reconstruction $\hat{x}$, foreground mask $F$ masking pixels with intensities above 0 and of application of median filter $f_{MF}$:

$$A(x) = f_{MF} \left( F \odot \sum_m^M \frac{|x_m - \hat{x}_m|}{M} \right).$$

No noise is used at test time.

## 4. Experiment Setup

### 4.1. Dataset

We evaluate the anomaly detection performance on the surrogate task of brain tumor segmentation using data from the BraTS 2021 challenge (Menze et al., 2014; Bakas et al., 2017, 2018). This data comprises native (T1), post-contrast T1-weighted (T1Gd), T2-weighted (T2), and T2 Fluid Attenuated Inversion Recovery (FLAIR) modality volumes for each patient from a variety of institutions and scanners. The data has already been co-registered, skull-stripped and interpolated to the same resolution. Labels are provided for tumor sub-regions: the GD-enhancing tumor, the peritumoral edema, and the necrotic and non-enhancing tumor.

We randomly split the dataset into 938 training, 62 validation, and 251 test patients. We consider the union of the tumor sub-region labels to be the anomalous regions. During training, we use only slices that do not contain any tumor pixels, under the assumption that these non-tumor slices represent healthy tissue. For the data input to the models, we concatenate all four modalities at the channel dimension for each patient. We normalize (rescale) the pixel values in each modality of each scan by dividing by the 99th percentile foreground voxel intensity. All slices are downsampled to a resolution of $128 \times 128$ (1.62mm/pixel).

### 4.2. Baselines

We compare the DAE anomaly detection performance against four baselines. Firstly, we implement a standard VAE (Zimmerer et al., 2019; Zhou et al., 2020) and f-AnoGAN (Schlegl et al., 2019) models with pixelwise reconstruction error as the anomaly scores. Secondly, we use the same VAE model but use an iterative gradient-based restoration process (You et al., 2019) to produce restoration images. Finally, we apply the simple thresholding approach from Meissen et al. (2021) using both their original post-processing procedure and our proposed modified procedure that includes median filtering. We use the hyperparameters from the original works for the deep learning methods but tune manually where necessary to improve training stability and AD performance.

### 4.3. Implementation details

**Denoising autoencoder** We use an encoder-decoder architecture with three downsampling/upsampling stages. Each encoder stage consists of two weight-standardized convolutions (Qiao et al., 2019) with kernel sizes of 3 and 64, 128, 256 output channels for the three stages respectively followed by Swish activations (Ramachandran et al., 2017) and group normalization (Wu and He, 2018). Average $2 \times 2$ pooling is used for downsampling. The decoder architecture mirrors the encoder in reverse, using transposed convolutional layers for upsampling. Architecture visualization and further details can be found in Appendix A.

Noise is generated by sampling random Gaussian pixelwise noise at the resolution of $16 \times 16$ pixels then bilinearly upsampled to the input resolution of $128 \times 128$ pixels. The generated noise is then randomly translated vertically and horizontally to randomize the centers of the coarse noise clusters that may occur due to upsampling from very low resolutions. Noise

is generated independently for each image modality. We investigate the parameters of the noise in Section 5 (see Figure 4).

We use mean $L2$ reconstruction loss in the foreground as the training objective. Models are trained for 67,200 iterations with a batch size of 16 slices using Adam (Reddi et al., 2018) with a cosine annealed maximum learning rate of 0.0001 with a period of 200 iterations.

DAE code is available at https://github.com/AntanasKascenas/DenoisingAE.

**VAE reconstruction** We use a similar architecture to train VAE models. Skip connections are removed and a bottleneck with dimensionality of 128 is added. We use the sum of mean $L2$ reconstruction error and KL-divergence with a weight of $\beta = 0.001$ as the training objective. We use the same training procedure and anomaly score formula as for the DAE.

**VAE restoration** Using the VAE model described above, we implement a restoration method (You et al., 2019) to produce the anomaly scores. We perform the restoration procedure using 100 iterations on individual slices basing our implementation on public source code [1]. Note that due to the iterative nature of the restoration procedure it takes significantly longer (approx. $\times 100$) to produce predictions compared to other methods.

**f-AnoGAN** We adapt the original public implementation [2] for the brain MR data task as follows. We use an additional generator, discriminator and encoder block to account for the higher resolution. Strided convolutions and transposed convolutions are used for downsampling and upsampling respectively. We use a batch size of 32 and learning rates of 0.001, 0.001, 0.00001 for the generator, discriminator and encoder respectively. The encoder was trained using $\kappa = 1 \times 10^{-8}$.

**Thresholding** We follow Meissen et al. (2021) to obtain results for the thresholding baseline. FLAIR volumes are histogram equalized in the foreground and connected component filtered to produce anomaly scores. We also experiment with using median filtering instead of connected component filtering to computationally simplify the baseline.

**Postprocessing** We experiment with applying the following postprocessing steps to our method and the baselines. We use a median filter with a kernel size of 5. Connected componenent filtering is done by discarding connected components with volume no larger than 20 voxels after binarization following Meissen et al. (2021). The postprocessing combination of median filtering in 3D and connected component filtering was used by Baur et al. (2021).

## 5. Results

We evaluate the anomaly detection performance of the methods with two metrics. Firstly, we measure the area under the precision-recall curve (AUPRC) at the pixel level computed for the whole test set. AUPRC evaluates anomaly scores directly without requiring to set an operating point for each method. Secondly, we calculate $\lceil$Dice$\rceil$, a Dice score which measures the segmentation quality using the optimal threshold for binarization found by sweeping over possible values using the test ground truth. $\lceil$Dice$\rceil$ represents the upper bound for the Dice scores that would be obtainable in a more practical scenario.

---

1. https://github.com/yousuhang/Unsupervised-Lesion-Detection-via-Image-Restoration-with-a-Normative-Prior
2. https://github.com/tSchlegl/f-AnoGAN

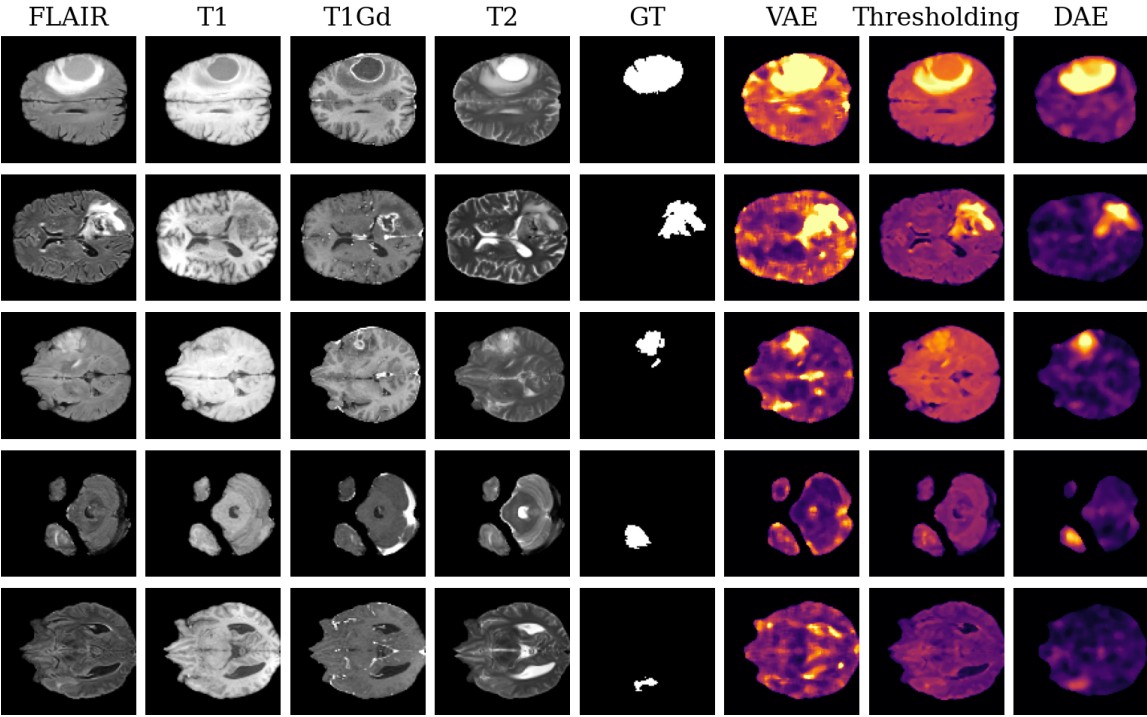

Figure 3: Sample anomaly score predictions. From easier (top) to more difficult (bottom).

Table 1: Tumor detection performance as evaluated by test set wide pixel-level area under the precision-recall curve (AUPRC) and ideal Dice score ($\lceil$Dice$\rceil$). MF refers to the application of median filtering in post-processing. CC refers to connected component filtering. $\pm$ indicates standard deviation across 3 runs.

| Method | AUPRC | $\lceil$Dice$\rceil$ | $\lceil$Dice$\rceil$ (+CC filtering) |
|---|---|---|---|
| Thresholding | 0.684 | 0.667 | 0.679 |
| Thresholding + MF | 0.798 | 0.749 | 0.750 |
| f-AnoGAN | $0.198_{\pm0.006}$ | $0.316_{\pm0.006}$ | $0.327_{\pm0.007}$ |
| f-AnoGAN + MF | $0.365_{\pm0.024}$ | $0.449_{\pm0.014}$ | $0.453_{\pm0.015}$ |
| VAE (reconstruction) | $0.299_{\pm0.002}$ | $0.395_{\pm0.002}$ | $0.405_{\pm0.002}$ |
| VAE (reconstruction) + MF | $0.555_{\pm0.004}$ | $0.548_{\pm0.003}$ | $0.551_{\pm0.003}$ |
| VAE (restoration) | $0.740_{\pm0.007}$ | $0.685_{\pm0.005}$ | $0.686_{\pm0.005}$ |
| VAE (restoration) + MF | $0.750_{\pm0.006}$ | $0.689_{\pm0.005}$ | $0.690_{\pm0.005}$ |
| DAE (ours) | $0.816_{\pm0.005}$ | $0.758_{\pm0.004}$ | $0.763_{\pm0.004}$ |
| DAE + MF (ours) | $\mathbf{0.833_{\pm0.005}}$ | $\mathbf{0.773_{\pm0.004}}$ | $\mathbf{0.774_{\pm0.004}}$ |

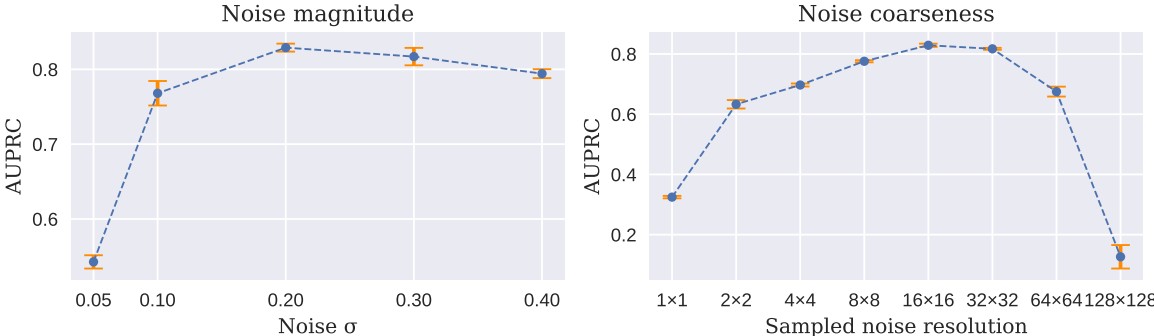

Figure 4: Noise coarseness and magnitude ablation results on validation data. Magnitude ablation uses noise sampled at resolution of 16×16. Coarseness ablation uses $\sigma = 0.2$. Error bars show standard deviation across three runs.

Quantitative results as evaluated using the BraTS 2021 dataset can be seen in Table 1. Some prediction samples are provided in Figure 3. Our simple DAE model outperforms f-AnoGAN (Schlegl et al., 2019), VAE reconstruction (Zimmerer et al., 2019; Zhou et al., 2020), VAE restoration (You et al., 2019) and thresholding (Meissen et al., 2021) baselines.

We also observe that adding median filtering improves the performance of all tested methods. We find it both more effective and computationally less demanding than connected component filtering at reducing false positive detections. Histogram-equalized and median-filtered image thresholding provides a strong baseline that challenges state-of-the-art anomaly detection methods in the brain tumor evaluation protocol.

To examine the effect of noise in DAEs we further investigate the effect of the sampled noise resolution before upsampling and the $\sigma$ of the Gaussian distribution used for sampling noise (see Figure 4 and Appendix C). We find that a reasonably coarse noise is critical, as DAE models trained using standard pixel-level noise (generated at $128 \times 128$ resolution) or using the opposite extreme of image-level noise (generated at $1 \times 1$ resolution) perform significantly worse. DAEs seems to be not so sensitive to the magnitude of the noise ($\sigma$ of the generating Gaussian distribution) however it can still have a significant effect on the results suggesting that further investigation into noise distributions could be fruitful.

## 6. Conclusion

In this paper we have proposed a method based on denoising autoencoders for anomaly detection, and presented an evaluation on brain tumor data. We found that a relatively simple DAE implementation with appropriate design of the noise can produce state-of-the-art results. Furthermore, we find that median filtering improves the results of both thresholding and DAE methods, proving to be a simple and effective post-processing step. In summary, we believe these methods provide strong baselines for future approaches in brain anomaly detection. However, it is possible that current evaluation protocols do not sufficiently assess anomalies with diverse intensity profiles and better evaluation datasets might be needed for progress towards more general anomaly detection methods.

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

## Appendix A. Neural network architectures

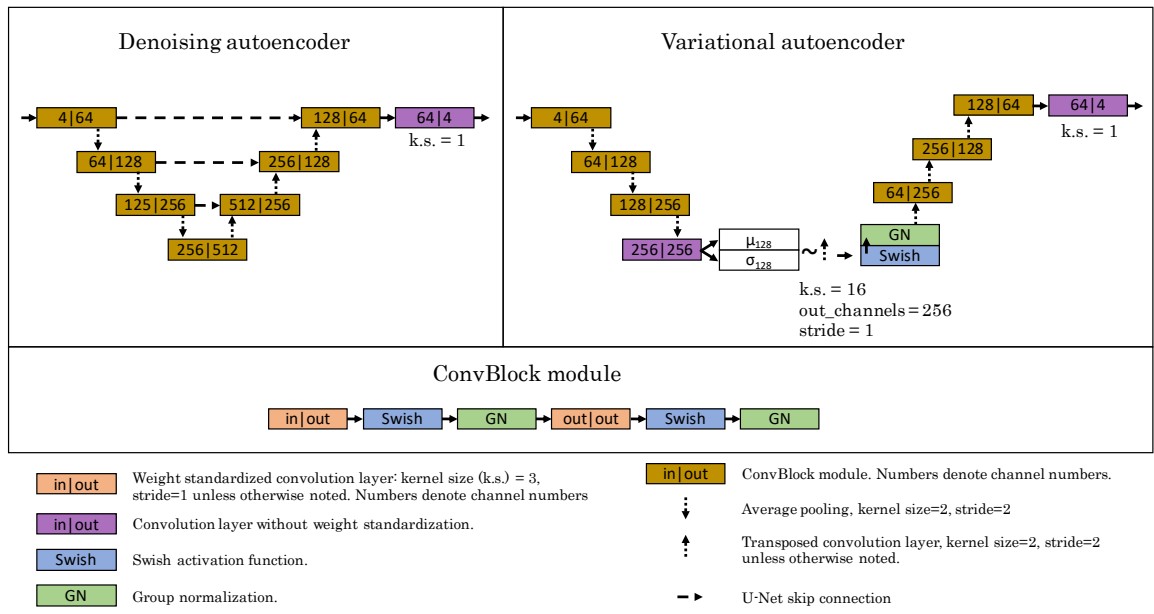

Figure 5: Architectures of DAE and VAE models used in the experiments.

## Appendix B. Noise samples

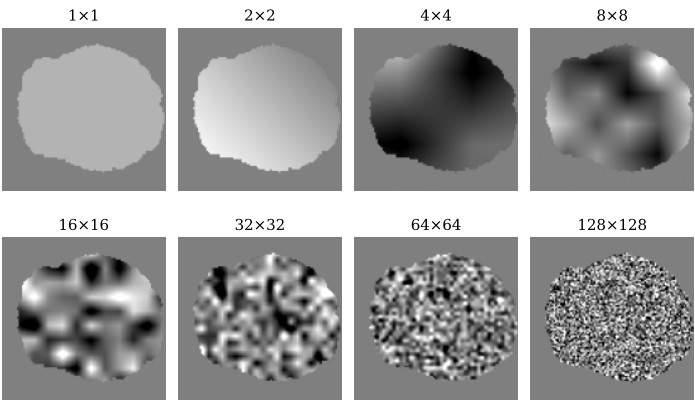

Figure 6: Samples of noise generated by bilinearly upsampling Gaussian pixelwise noise using different initial resolutions, from 1×1 through to 128×128.

## Appendix C.  Noise generation ablation results

Table 2:  Noise resolution ablation on validation data.  Noise is generated at the specified resolution and then upsampled to the image resolution of $128{\times}128$.  Thus, $128{\times}128$ refers to pixelwise noise and $1{\times}1$ refers to image level noise.  Noise is generated with $\mathcal{N}(0,\,0.2^2)$

.

| Method | AUPRC | $\lceil$Dice$\rceil$ |
|---|---|---|
| DAE ($1 \times 1$ noise) | $0.325_{\pm0.004}$ | $0.354_{\pm0.011}$ |
| DAE ($2 \times 2$ noise) | $0.633_{\pm0.014}$ | $0.589_{\pm0.009}$ |
| DAE ($4 \times 4$ noise) | $0.697_{\pm0.005}$ | $0.634_{\pm0.006}$ |
| DAE ($8 \times 8$ noise) | $0.776_{\pm0.004}$ | $0.711_{\pm0.004}$ |
| DAE ($16 \times 16$ noise) | $\mathbf{0.829_{\pm0.005}}$ | $\mathbf{0.765_{\pm0.005}}$ |
| DAE ($32 \times 32$ noise) | $0.817_{\pm0.003}$ | $0.755_{\pm0.004}$ |
| DAE ($64 \times 64$ noise) | $0.675_{\pm0.016}$ | $0.637_{\pm0.006}$ |
| DAE ($128 \times 128$ noise) | $0.127_{\pm0.040}$ | $0.186_{\pm0.045}$ |

Table 3:  Noise magnitude ablation on validation data.  Noise is generated at a resolution of $16 \times 16$ using $\mathcal{N}(0,\,\sigma^2)$ with the specified $\sigma$.

| Method | AUPRC | $\lceil$Dice$\rceil$ |
|---|---|---|
| DAE (noise $\sigma = 0.05$) | $0.543_{\pm0.009}$ | $0.582_{\pm0.006}$ |
| DAE (noise $\sigma = 0.1$) | $0.768_{\pm0.016}$ | $0.715_{\pm0.014}$ |
| DAE (noise $\sigma = 0.2$) | $\mathbf{0.829_{\pm0.005}}$ | $\mathbf{0.765_{\pm0.005}}$ |
| DAE (noise $\sigma = 0.3$) | $0.817_{\pm0.012}$ | $0.752_{\pm0.011}$ |
| DAE (noise $\sigma = 0.4$) | $0.794_{\pm0.006}$ | $0.729_{\pm0.004}$ |

