# OpenReview forum: "Denoising Autoencoders for Unsupervised Anomaly Detection in Brain MRI"
_MIDL.io/2022/Conference — MIDL 2022_

### Official Review · Reviewer_LKYe · 2022-01-24

**Confidence:** 4
**Preliminary Rating:** 4
**Recommendation:** Poster

**Summary:**

This paper uses denoising autoencoders for unsupervised anomaly detection in brain MRI scans. A noise generation process is proposed that allows to tune the spatial scale and magnitude of noise patterns. In experiments on a brain tumor dataset, the authors demonstrate that their method can outperform several baselines when the noise scale and magnitude are chosen appropriately. As opposed to several previous methods, their DAE-based approach outperforms a simple thresholding-based baseline, which could be of interest to the community.

**Strengths:**

Although (as the authors state) DAEs have been used before in the context of anomaly detection, they did not receive as much attention as alternative models like VAEs or GANs. The authors are the first to propose a UAD method based on DAEs that improves upon multiple existing baselines through simple modifications of the noise generation process. This result is potentially interesting for the community. Apart from that, the manuscript shows that a different post-processing further improves a previously published thresholding baseline, which remains relevant in future UAD papers with brain MRIs. In general, the paper is well-written and easy to follow.

**Weaknesses:**

The proposed method is evaluated only on a single dataset, which is not enough to judge the generality of an unsupervised anomaly detection (UAD) method. While the results showcase the potential of DAEs in UAD and are thus still interesting for the community in my opinion, adding another dataset as done in many related works (e.g. MS lesion datasets to see if the proposed method works with smaller anomalies, too) would be important to determine how practical the method is. After all, the goal of UAD methods is to detect *multiple* types of anomalies after training on a healthy cohort, so hyperparameter settings tuned on a single anomaly type annotated in the validation set may not work for other anomalies.

Apart from the question whether the hyperparameters chosen with the annotated validation set generalize to other datasets/anomalies, I think the comparison with baselines could be made more transparent by describing the hyperparameter selection and tuning budget more clearly. Was each baseline tuned manually or values adopted from related work?

**Deanonymize Review:**

yes

**Detailed Comments:**

Methods:
- Please specify the median filter size used in your experiments (here or in implementation details).
- (minor) Maybe note that a post-processing combining brain mask, median filter and CC has been used before (e.g. Baur 2021).
- (minor) fig. 2 could be made smaller by only showing FLAIR; maybe instead of multiple modalities, more baseline reconstructions would be interesting?

Experiment setup:
- Cosine lr-schedule is a rather uncommon choice (if I understand correctly that the learning rate follows the cosine with periodicity of 200 steps). Could you provide some motivation for this choice?
- (minor) CC should be defined as a general post-processing method, since it is applied to all baselines. Currently it is only found in the thresholding paragraph.

Results:
- (minor) Last column of table 1 (dice + CC) is not a different metric, but a different post-processing method. Although it is not really confusing, the authors may consider separating their comparison of post-processing methods from the other results.
- (minor) If a full grid-search over noise magnitude and resolution was done, the results could be added to the appendix, instead of the tables that contain the same information as the figure.


**Paper Type:**

methodological development

**Questions To Address In The Rebuttal:**

In my opinion, the main weakness of this paper is the evaluation on just one dataset, so adding results on another dataset could still change my rating. However, I understand that adding new experiments is most likely out of scope for the rebuttal.
The authors should still clarify the hyperparameter tuning method and some individual hyperparameter choices (see detailed comments).

**Special Issue:**

no

---

### Official Review · Reviewer_H5GP · 2022-01-24

**Confidence:** 4
**Preliminary Rating:** 4
**Recommendation:** Oral

**Summary:**

Unsupervised anomaly detection (UAD) has been widely studied and most successfully using variational autoencoders (VAEs). In this work, the authors address the poor reconstruction problems with information bottleneck methods such as VAEs by using a denoising autoencoder (DAE). Random, Gaussian noise masks are added to each modality of multi-modal input data and reconstruction task is set up as denoising. Thorough evaluations on public dataset and relevant baseline methods show that DAEs implemented using U-net like architectures are better at UAD.

**Strengths:**

* The focus on improving reconstructions to help UAD is a simple and logical one. The information bottlenecks in VAEs have known to affect reconstructions, and hence also UAD. By focusing on using skip connections (U-net) and designing suitable noise, this work shows promising results.

* The paper is well written, with a balanced overview of relevant literature.

* Baseline experiments are shown with appropriate methods, and the performance improvements are considerable.

* Noise generation model is simple and effective; the study showing the influence of noise magnitude and coarseness provides additional insight.

**Weaknesses:**

* The simplicity of the nosie model is actually a strong point in this work. However, one can't but wonder if there would be more gains if a well-suited noise model can be used for this work? Have the authors considered other noise models that perhaps take the spatial information into account? Or on the other extreme would these results hold for noise models used in self-supervised learning, such as random masking.

* How are the optimal thresholds obtained for Dice computation? Is the test set used for choosing this threshold as mentioned here:
> Secondly, we calculate ⌈Dice⌉, a Dice score which measures the segmentation quality using the optimal threshold for binarization found by sweeping over possible values using the test ground truth.
>

* Will the same denoising strategy work for other types of anomaly detection? BraTS dataset is challenging but as the experiments also show, thresholding + MF do sufficiently well. Could this be the case because of the multi-modality nature of the dataset?

* Source code is for the work is not available. I encourage the authors to provide a repository so that the results can be reproduced.

**Deanonymize Review:**

no

**Detailed Comments:**

See points above.

**Final Rating After The Rebuttal:**

4: Weak Accept

**Justification Of The Final Rating:**

Appreciate the responses from the authors and for making the source code available. The discussions about the use of test set for obtaining optimal thresholds were not sufficiently addressed here, but the discussion with Reviewer FWoZ was interesting. It is a bit peculiar that UAD literature prefers to report the upper bound Dice using the test set. But like the authors said, it might be for another discussion. Finally, as the work originally is validated on single dataset, I will keep the score at Weak Accept.

**Paper Type:**

both

**Questions To Address In The Rebuttal:**

* Discussion on what the influence of other noise models might be could be interesting
* Discussion on how this method could work on other datasets can be useful.
* Source code could be made available available

**Special Issue:**

yes

---

### Official Review · Reviewer_FWoZ · 2022-01-26

**Confidence:** 3
**Preliminary Rating:** 4
**Recommendation:** Poster

**Summary:**

This paper investigates a simple denoising auto-encoder baseline for unsupervised anomaly detection, compared to the (popular) variational auto-encoder methods.

The results show that a modified u-net for reconstruction, trained with gaussian noise on the input, out-perform more complex methods.


---
I am limited in the auto-encoder literature, so I do not score my confidence high and I grade the paper in a conservative way (weak accept).

**Strengths:**

- Good results with a simple baseline, compare to several relevant existing works
- Seemingly good overview of the current litterature (I am not knowledgeable in neither auto-encoder nor unsupervised anomaly detection)

**Weaknesses:**

The inference and post-processing section (3.3) is way to short and does not allow to fully understand the in and out of the supervision. For instance, I am confused by the "foreground mask $F$", at inference. What is this? How does $f_{MF}$ work?

I also think that the authors could motivate a bit more some of the design choices, such as the training regiment and the choice not to mix different noise scales.

I would also have liked to see a discussion on the (total) training time, training stability, and inference, of the different methods.

**Deanonymize Review:**

yes

**Detailed Comments:**

I would encourage the authors to make their code available to other researchers, even more so as they relied on two public code for their baselines; it is important to "give back".


Minor:
- there is a trailing "D" in Table 3 (third line)
- Could use `\left(` and `\right)` parenthesis for $A(x)$, i.e.: $A(x) = f_{MF}\left( F \odot \sum_m^M \frac{|x_m - \hat x_m}{M}\right)$

**Final Rating After The Rebuttal:**

5: Strong Accept

**Justification Of The Final Rating:**

I am happy with the response to all reviewers, and I have no major concern about the paper. I recommend a poster, and to **consider the paper for the MedIA special issue**.

Reviewer H5GP rightfully reminds that the current work is evaluated on a single dataset, which could be a weakness. To me, this is fine *for the conference*, but a journal extension will indeed require a much more thorough and diverse evaluation.

**Paper Type:**

both

**Questions To Address In The Rebuttal:**

- Why not mixing different coarseness of noise, during training?
- How is $f_{MF}$ computed?
- What is $F$ exactly?
- Why not a k-fold cross-validation? How the dataset was split initially ? (randomly or sorted in some way)
- Are some methods more prone to overfitting or more sensitive to initial (random) weights?

**Special Issue:**

no

---

### Meta-Review · Area_Chair_4Tqj · 2022-02-14

**Recommendation:** Accept (Poster)
**Confidence:** 5

**Metareview:**

This paper presents an in-depth analysis of denoising autoencoders for anomaly detection. The reviewers raised some initial questions in their comments, which were mostly addressed by the authors in their rebuttal. All reviewers now agree that the paper is ready for publication at MIDL. Based on their recommendation, I'm happy to accept this work.

---

### Decision · Program_Chairs · 2022-02-28

Accept